# Slimming the Grain through Breeding Is a Practical Way to Reduce the Chalky Grain Rate of Middle-Season Hybrid Rice

**Min Huang *** , **Jialin Cao, Jiana Chen, Fangbo Cao and Chuanming Zhou**

Rice and Product Ecophysiology, Key Laboratory of Ministry of Education for Crop Physiology and Molecular Biology, Hunan Agricultural University, Changsha 410128, China
* Correspondence: mhuang@hunau.edu.cn

**Abstract:** The production of middle-season rice is an important part of agriculture in the Yangtze River basin of China. In recent years, the chalky grain rate of middle-season rice has decreased with the release of new cultivars. However, limited information is available on the factors responsible for this change in the chalky grain rate. This study evaluated the trends in the chalky grain rate and grain size traits of the new cultivars and the relationships between the chalky grain rate and grain size traits for middle-season hybrid rice in a province located in the middle reaches of the Yangtze River basin during 2006–2021. The results indicate that the recently reduced chalky grain rate of middle-season hybrid rice in the new cultivars is closely associated with a decrease in rice width, suggesting that it is feasible to reduce the chalky grain rate of middle-season hybrid rice by slimming the grain through breeding.

**Keywords:** chalkiness; grain size; indirect breeding; rice

## 1. Introduction

Rice is grown in almost all provinces in China [1] and is consumed by approximately 70% of the population as a staple food [2]. Rice-based cropping systems are diverse in China due to the existence of various agroclimatic zones [3]. Middle-season rice is an important part of rice production in the Yangtze River basin [4] and is always encouraged to be grown in rotation with winter crops (e.g., oilseed rape and wheat) in regions where thermal energy is not sufficient for successively growing early- and late-season rice within a single calendar year (i.e., double-season rice).

In recent years, the planting area for middle-season rice has sharply increased in the major double-season rice-producing region, mainly because of a reduction in rural labor in addition to rising labor wages due to urbanization and economic growth [5]. In conjunction with the increased planting area of middle-season rice, highly-efficient ratoon rice cropping systems with middle-season rice grown from April to August as the main crop have become more and more widespread in the Yangtze River basin [6].

Another important reason for the rapid development of middle-season rice in the Yangtze River basin, especially ratoon rice, is that multiple middle-season rice cultivars with superior quality grains have been developed in recent years [7,8]. These new cultivars can meet the increased demand for and consumption of high-quality rice in China due to improved economic and living standards [9]. The successful breeding of high-quality middle-season rice cultivars also contradicts the previous view that high grain quality is difficult to achieve for main crops in ratoon rice cropping systems because they generally experience high temperatures during the ripening period (July to August). For example, the average daily mean temperature during July to August was 2–6 °C higher than the upper limit of the optimum daily mean temperature for rice ripening (25 °C) during 2011–2021 in Hunan Province, located in the middle reaches of the Yangtze River basin (Figure 1).

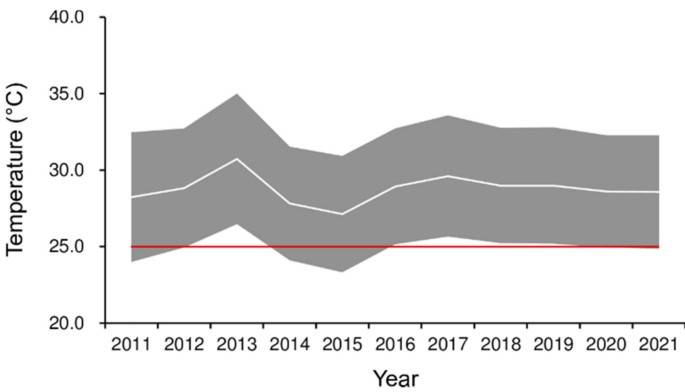

**Figure 1.** Average daily mean (white line within the gray area), maximum (upper edge of the gray area), and minimum (lower edge of the gray area) temperatures during July and August in Hunan Province of China from 2011 to 2021. Data are averaged across 14 cities of Hunan Province, including Changde, Changsha, Chenzhou, Hengyang, Huaihua, Jishou, Loudi, Shaoyang, Xiangtan, Yiyang, Yongzhou, Yueyang, Zhangjiajie, and Zhuzhou (Supplementary Spreadsheet S1). The horizontal red line represents the upper limit of the optimum daily mean temperature for rice ripening (25 °C) [10].

Chalkiness (i.e., opacity in the endosperm) is an appearance trait of rice that affects customer selection when buying rice, thus defining the market value of rice [11]. In recent years, the chalky grain rate of rice in China, including middle-season rice in the Yangtze River basin, has decreased with the release of new cultivars [12,13]. A better understanding of this change has important implications for developing new rice cultivars with a low chalky grain rate.

Rice chalkiness is a complex polygenic quantitative trait, and many quantitative trait loci (QTLs) that are associated with chalkiness have been identified [14]. However, because almost all the detected QTLs associated with chalkiness have not been practically used in recent breeding programs, these QTLs cannot be used directly to explain the change in chalky grain rate with the development of new rice cultivars.

Grain chalkiness is associated with many other grain traits in rice. More specifically, the chalky grain rate generally decreases with a decrease in the grain size of rice [15,16]. Recently, several QTLs have been identified with pleiotropic effects on the chalky grain rate and grain size [17,18]. These suggest that a lower chalky grain rate can be indirectly achieved by decreasing the grain size of rice. However, it is not clear whether this indirect path is responsible for the recently reduced chalky grain rate of the new rice cultivars.

To address this knowledge gap, we evaluated (1) the trends of the chalky grain rate and the grain size traits of the new cultivars and (2) the relationships between the chalky grain rate and the grain size traits.

## 2. Materials and Methods

### 2.1. Data Collection

We collected data on the chalky grain rate and grain size traits (grain weight as well as rice length, width, and length/width ratio) of the middle-season hybrid rice cultivars released in Hunan Province during 2006–2021 from the China Rice Data Center [19]. These data were obtained from the regional trial of rice cultivars in Hunan Province. In brief, tested rice cultivars were grown in a randomized block design with three replications and a plot size of 13.3 m$^2$. Seeds were sown on 20–25 April. Seedlings were transplanted at a hill spacing of 20.0 cm × 26.7 cm, with two seedlings per hill. Crop management was performed according to locally recommended practices. Three plants were sampled and hand-threshed, and two subsamples of 1000 filled grains were used to determine the grain weight. The chalky grain rate, rice length, and length/width ratio were measured by the Food Quality Supervision and Testing Center of the Ministry of Agriculture and Rural

Affairs of China (Wuhan). The rice width was calculated by dividing the rice length by the rice length/width ratio. All data are provided in Supplementary Spreadsheet S2.

*2.2. Statistical Analysis*

Linear regression analysis was employed to evaluate the time trends in changes of the chalky grain rate and grain size traits and the relationships between the chalky grain rate and grain size traits and between the grain size traits (Statistix 8.0, Analytical Software, Tallahassee, FL, USA).

### 3. Results and Discussion

The chalky grain rate decreased significantly with the year of cultivar release (Figure 2a). There were several cultivars released in 2021 (e.g., Yunliangyou 526, Zhenliangyouyuzhan, and Hongliangyou 2137) having a chalky grain rate of ≤10% (Supplementary Spreadsheet S2), meeting the national standard of first-grade high-quality rice in China [20]. The grain weight experienced a significant decrease with the year of cultivar release (Figure 2b). The rice length did not change significantly, while the rice width decreased significantly with the year of cultivar release (Figure 2c,d). The rice length/width ratio increased significantly with the year of cultivar release (Figure 2e). The decreased rice width and increased rice length/width ratio with the development of new rice cultivars were mainly attributed to the recently increased demand for high-quality slender-grain rice (length/width ratio ≥ 3), which has led breeders to develop rice cultivars with low rice width in China [21].

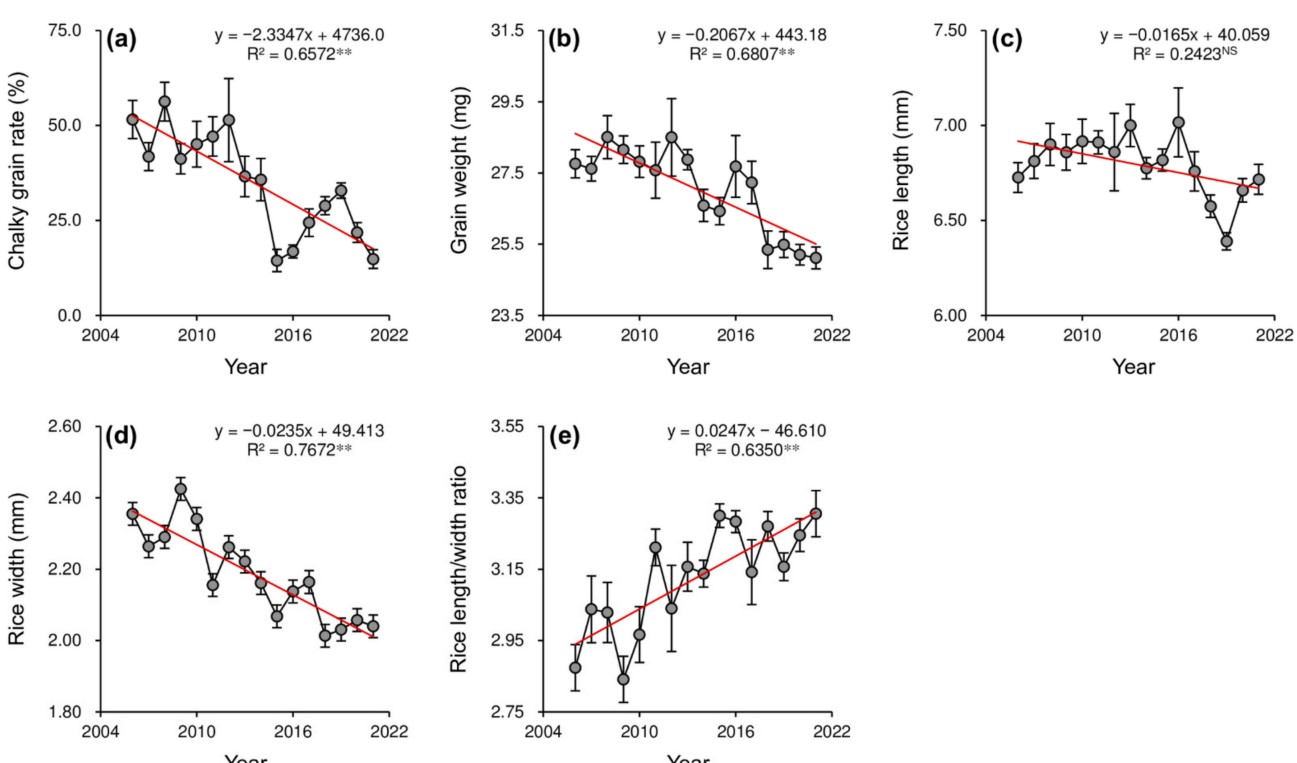

**Figure 2.** Trends in the chalky grain rate (**a**), grain weight (**b**), rice length (**c**), rice width (**d**), and rice length/width ratio (**e**) of middle-season hybrid rice cultivars released in Hunan Province of China from 2006 to 2021. Data are presented as the means ± SE ($n$ = 5–30). $^{NS}$ denotes a nonsignificant trend, $p > 0.05$. ** denotes a significant trend, $p < 0.01$.

A significantly negative relationship was observed between the chalky grain rate and the grain weight (Figure 3a). The relationship between the chalky grain rate and the grain length was not significant (Figure 3b). The chalky grain rate was significantly positively

related to the rice width and significantly negatively related to the rice length/width ratio (Figure 3c,d).

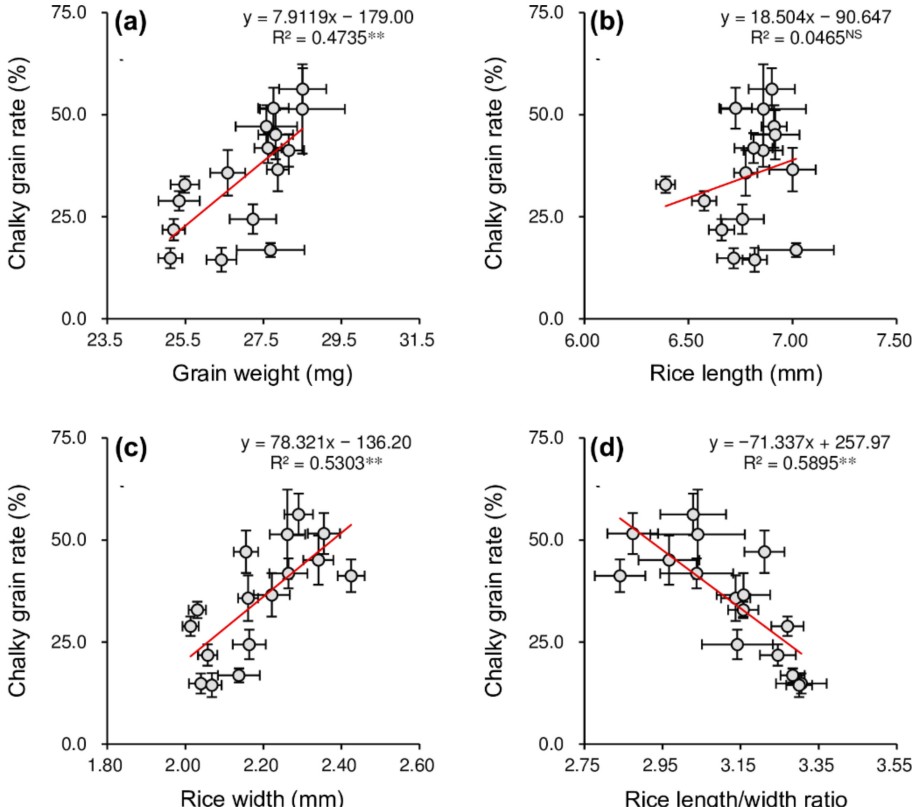

**Figure 3.** Relationships between the chalky grain rate and the grain weight (**a**), rice length (**b**), rice width (**c**), and rice length/width ratio (**d**) across middle-season hybrid rice cultivars released during 2006–2021 in Hunan Province of China. Data are presented as the means ± SE (*n* = 5–30). $^{NS}$ denotes a nonsignificant relationship, *p* > 0.05. ** denotes a significant relationship, *p* < 0.01.

There were significantly positive relationships between the grain weight and both the rice length and width (Figure 4a,b). The grain weight was more closely related to the rice width than to the rice length: approximately 56% and 72% of the variation in the grain weight was explained by the rice length and width, respectively. The rice length/width ratio was not significantly related to the rice length but was significantly negatively related to the rice width (Figure 4c,d). The rice width explained 85% of the variation in the grain weight.

Taken together, the results of this study indicate that the recently reduced chalky grain rate of middle-season hybrid rice with the development of new cultivars is closely associated with a decrease in the rice width. In this regard, it has been well-documented that the trait of chalkiness is easily generated during the grain-filling process in large and round rice grains, due to the long-distance transportation of assimilates from the back to the belly of the grain [22]. The finding of this study suggests that slimming the grain through breeding, either conventional breeding or genetic engineering, is a practical approach to reducing the chalky grain rate of middle-season hybrid rice.

As grain size is a yield determinant in rice, there may be some concern that a decrease in rice width can result in a decrease in grain weight, which can consequently cause a reduction in yield. However, our study showed that a recently developed high-quality hybrid rice cultivar, Jingliangyou 1468, had a 13% lower grain weight but produced a 15% higher grain yield than an older high-yield hybrid rice cultivar, Liangyoupeijiu [8]. The higher grain yield in Jingliangyou 1468 was mainly due to a higher growth rate during the post-heading period, resulting in a higher harvest index and hence a higher spikelet filling percentage compared to Liangyoupeijiu. Therefore, developing cultivars with a small

grain size and late-stage vigor may be a feasible way to achieve a compatible relationship between the grain yield and the appearance quality of rice.

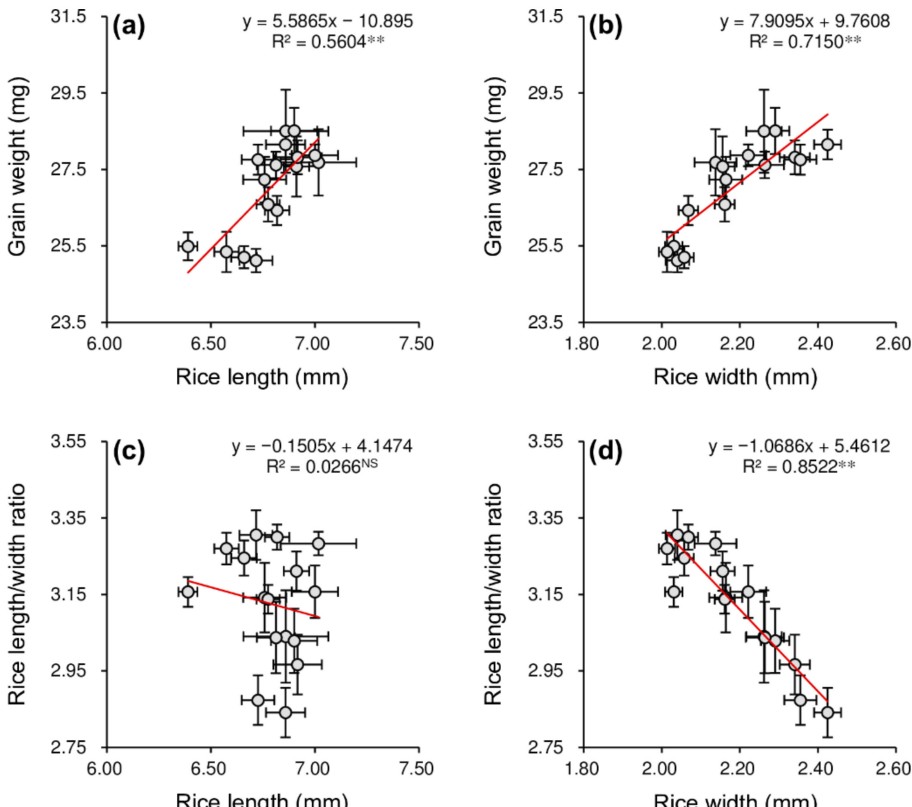

**Figure 4.** Relationships between the grain weight and rice length (**a**) and rice width (**b**), as well as rice length/width ratio and rice length (**c**) and rice width (**d**) across middle-season hybrid rice cultivars released during 2006–2021 in Hunan Province of China. Data are presented as the means ± SE (*n* = 5–30). $^{NS}$ denotes a nonsignificant relationship, $p > 0.05$. ** denotes a significant relationship, $p < 0.01$.

Nevertheless, this study had certain limitations that should be addressed in further studies. First, the present study does not provide direct evidence that slimming the grain can reduce the occurrence of chalkiness in rice under high temperature conditions. Second, this study does not evaluate the effects of slimming the grain on other grain quality traits such as milling recovery and palatability. Therefore, further investigations are required to (1) determine the responses of the chalky grain rate to high temperature in rice cultivars differing in grain width and (2) comprehensively assess the feasibility of the strategy of slimming the grain for the improvement of rice quality.

**Supplementary Materials:** The following supporting information can be downloaded at: https://www.mdpi.com/article/10.3390/agronomy12081886/s1, Spreadsheet S1: Average daily mean, maximum, and minimum temperatures in July and August in 14 cities of Hunan Province of China during 2011–2021. Spreadsheet S2: The chalky grain rate and grain size traits (grain weight as well as rice length, width, length, and length/width ratio) of middle-season hybrid rice cultivars released in Hunan Province during 2006–2021.

**Author Contributions:** Conceptualization, M.H.; investigation, J.C. (Jialin Cao), J.C. (Jiana Chen), F.C. and C.Z.; writing—original draft preparation, M.H.; funding acquisition, M.H. All authors have read and agreed to the published version of the manuscript.

**Funding:** This research was funded by the National Key R&D Program of China, grant number 2016YFD0300509.

**Institutional Review Board Statement:** Not applicable.

**Informed Consent Statement:** Not applicable.

**Data Availability Statement:** The data presented in this study are available in the Supplementary Materials.

**Conflicts of Interest:** The authors declare no conflict of interest.

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
