# Peer review of "Slimming the Grain through Breeding Is a Practical Way to Reduce the Chalky Grain Rate of Middle-Season Hybrid Rice"

_agronomy, doi:10.3390/agronomy12081886_

Round 1
Reviewer 1 Report
This is an interesting paper, as chalky rice is important to farmers, millers, and rice eaters. However, as a short communication, it is difficult to evaluate the scientific soundness of the paper as there is no description of the field methods, cultivar names, or statistical methods of analysis. Also, a good deal is known about the relationship between rice chalkiness and milling yield. Starch structural differences are also known yet no mention of this was made.
Chalky rice has been shown to be associated with high temperatures during grain development. Yet the authors don't address this issue. Only mean temperatures across many growing regions are reported. There is no statistical analysis to evaluate if the differences in grain chalkiness were related to temperatures during grain development. This must be addressed and the appropriate data included and discussed. If this isn't done, then at the least the authors need to address the fact that their study is based on correlations and thus the results must be considered with caution.
The paper needs to be divided into sections for ease of reading.
Author Response
This is an interesting paper, as chalky rice is important to farmers, millers, and rice eaters. However, as a short communication, it is difficult to evaluate the scientific soundness of the paper as there is no description of the field methods, cultivar names, or statistical methods of analysis. Also, a good deal is known about the relationship between rice chalkiness and milling yield. Starch structural differences are also known yet no mention of this was made.
Answer: (1) A brief description of the field methods has been added (Lines 77-90).
(2) Cultivar names are provided in Supplementary Spreadsheet S2.
(3) A sub-section of statistical analysis has been added (Lines 91-95).
(4) A paragraph has been added to acknowledge the limitations of this study (Lines 151-158).
Chalky rice has been shown to be associated with high temperatures during grain development. Yet the authors don't address this issue. Only mean temperatures across many growing regions are reported. There is no statistical analysis to evaluate if the differences in grain chalkiness were related to temperatures during grain development. This must be addressed and the appropriate data included and discussed. If this isn't done, then at the least the authors need to address the fact that their study is based on correlations and thus the results must be considered with caution.
Answer: We do not have the accurate data to evaluate the relationships between grain traits and temperature. We have acknowledged this limitation in the revised manuscript (Lines 151-154, 155-157).
The paper needs to be divided into sections for ease of reading.
Answer: The paper has been divided in three sections (Lines 21, 76, 96).
Reviewer 2 Report
Dear Authors,
The review report sounds promising. High-temperature results in chalkiness in rice grain. Have you intended to determine the relationships between temperature and grain chalkiness, grain width, or grain length? if yes, this result can be added to the report. it's just a suggestion.
Line 75: Please mention the reference of the database.
Author Response
Dear Authors,
The review report sounds promising. High-temperature results in chalkiness in rice grain. Have you intended to determine the relationships between temperature and grain chalkiness, grain width, or grain length? if yes, this result can be added to the report. it's just a suggestion.
Answer: Thank you very much for your good suggestion. However, we are sorry that we do not have the data to evaluate the relationships between grain traits and temperature. We have acknowledged this limitation in the revised manuscript (Lines 133-136, 137-139).
Line 75: Please mention the reference of the database.
Answer: A refence of the database has been added (Line 199).
Reviewer 3 Report
This manuscript will be better if you add more information about:
-The rice market standards in China about how slim the rice grains (related to grain width, grain length, grain length/width ratio) and chalky grain rate (%) that rice consumers can accept.
-Strategies or techniques to develop rice varieties with slim grain size and low chalky grain rate.
-Structure of the starch (measured by Scanning Electron Microscope) in slimmer rice grains and wider rice grains that growing under heat stress temperature. What happen if the slimmer rice grains showed similar starch structure with wider rice grains, is it will give the same rice taste between slimmer and wider rice grains? or the slimmer rice grains will give better rice taste than wider rice grains.
-Relationship between rice grain size with broken rice after milling process. Are slimmer rice grains that growing under heat stress temperature showed less broken rice after milling process? Or Wider rice grains that growing under heat stress temperature showed more broken rice after milling process?
Author Response
This manuscript will be better if you add more information about:
-The rice market standards in China about how slim the rice grains (related to grain width, grain length, grain length/width ratio) and chalky grain rate (%) that rice consumers can accept.
Answer: The China National Standard of the chalky grain rate for the first-grade high-quality rice is ≤ 10%. The slender-grain rice with a length/width ratio ≥ 3 is popular in southern China. These have been added in the revised manuscript (Lines 79-82, 88).
-Strategies or techniques to develop rice varieties with slim grain size and low chalky grain rate.
Answer: Either conventional breeding or genetic engineering can be used. This information has been added in the revised manuscript (Line 111).
-Structure of the starch (measured by Scanning Electron Microscope) in slimmer rice grains and wider rice grains that growing under heat stress temperature. What happen if the slimmer rice grains showed similar starch structure with wider rice grains, is it will give the same rice taste between slimmer and wider rice grains? or the slimmer rice grains will give better rice taste than wider rice grains.
-Relationship between rice grain size with broken rice after milling process. Are slimmer rice grains that growing under heat stress temperature showed less broken rice after milling process? Or Wider rice grains that growing under heat stress temperature showed more broken rice after milling process?
Answer: This study does not evaluate the effects of slimming the grain on other grain quality traits such as milling recovery and palatability. We have acknowledged this limitation in the revised manuscript (Lines 136-140).